# The role of external power demand on the choice of technique in classic cross-country skiing

**Gertjan Ettema** [ORCID]*, **Knut Skovereng, Tobias Ritman, Jørgen Danielsen**

Centre for Elite Sports Research, Department of Neuromedicine and Movement Science, Faculty of Medicine and Health Sciences, Norwegian University of Science and Technology, Trondheim, Norway

* gertjan.ettema@ntnu.no

**Data Availability Statement:** All relevant data are within the paper and its Supporting information files.

**Funding:** The author(s) received no specific funding for this work.

## Abstract

In cross-country skiing, athletes use different techniques akin to locomotor gaits such as walking and running. Transitions between these techniques generally depend on speed and incline, in a similar way as walk-run transitions. Previous studies have examined the roles of incline, speed, and mechanical power demand in triggering transitions. However, it is still not known if mechanical power demand, as an isolated factor, has any role on the choice of technique. The aim of this study was to examine the isolated role of mechanical power on the choice of technique during classic cross-country roller skiing by changing mechanical power demand at fixed speeds and inclines. Six male and eight female athletes performed classical roller skiing on a treadmill at the four combinations of two speeds (10 and 12 km h$^{-1}$) and two inclines (5 and 8%) while additional resistive forces were applied via a weight-pulley system. Athletes were free to choose between three techniques: double poling, double poling with kick, and diagonal stride. Power and resistive forces at transition were compared using repeated measure (2x2) ANOVA. At a given incline, technique transitions occurred at similar additional resistive force magnitudes at the two speeds. On the steeper incline, the transitions occurred at smaller additional resistive forces. Importantly, transitions were not triggered at similar mechanical power demands across the different incline/speed/resistive force conditions. This suggests that mechanical power itself is not a key technique transition trigger. Both total and additional resistive force (i.e., the manipulated mechanism to regulate power) may be transition triggers when incline is fixed and speed is changed. In combination with previous findings, the current results suggest that no single factor triggers technique transitions in classic cross-country skiing.

## Introduction

When increasing speed of locomotion, humans and animals spontaneously make a transition from one particular gait to another, e.g., walk to run [1–4] or trot to gallop [5]. Furthermore, the walk-run and reverse run-walk transitions typically occur at different speeds [e.g., 4, 6, 7], which is referred to as hysteresis. These phenomena are well-studied [e.g., 1, 2, 4] and there is

**Competing interests:** The authors have declared that no competing interests exist.

general consensus that these transitions strongly relate to metabolic cost [e.g., 3, 5] but also to mechanics and energetics [see e.g., 4, 6, 7]. Yet, while minimizing metabolic cost may be regarded a desired outcome for the choice of gait, the main (instantaneous) 'trigger' for transitions to occur, if one exists, has been hard to disentangle. For human walk-run transitions [e.g., 4, 6, 7], kinetic variables (peak forces, loading rates, muscle force-velocity) have been suggested as key determinants [4].

A similar challenge exists for understanding the technique transitions in cross-country skiing (XCS) [8, 9]. In classic style XCS, skiers make transitions (often dozens of times per race) between three main techniques: double poling (DP), double poling with kick (DK) and diagonal stride (DS) [10], locomotion techniques that have similarities with walking and running [11]. DP involves synchronous and symmetrical poling actions with propulsion generated solely through the poles (the skis remain gliding). DP is usually chosen on what is regarded as easy terrain, where speed tends to be fast. Kinematically, DS is similar to walking in terms of symmetrical but reciprocal arm and leg movements, with both skis and poles transmitting propulsive forces. DS is usually chosen on uphill terrain when speeds are slow. DK is similar to DP but with the addition of one ski thrust per poling cycle [4]. Although it may seem that the choice of and transition between techniques primarily depend on speed [e.g., 12–14], these studies propose other factors as well, and recent studies indicate that incline may be the more influential parameter [e.g., 8, 9, 15]. Incline, speed, and mechanical power are tightly coupled. Average external mechanical power is equal to the product of velocity and the sum of the resistive forces (i.e., gravity and frictional forces). Thus, even though techniques chosen are generally linked to incline, it is not apparent if one factor (incline, speed, or power) actually triggers the spontaneous transitions between techniques.

For classic style XCS, Pellegrini, Zoppirolli [16] provided data which supported the notion that a threshold in poling force exists so that DS (low pole forces) are preferred over DP (high pole forces) on steep uphill terrain at slow speeds. This idea was challenged by Dahl, Sandbakk [17], who suggested that on steep uphills, the mechanism that allows for considerable leg power contribution via utilization of the body's gravitational potential energy in pole propulsion is hampered, thereby disfavouring DP [15, 17]. Later, Danielsen, Sandbakk [18] showed that this mechanism, albeit less effective, was still very much present on steep inclines. They [18] further suggested that, because of the longer poling times on steep inclines and at slower speeds, the requirement to generate pole (and arm muscle) forces over a longer time may be a main factor for disfavouring DP. For DS, a limited leg thrust time in DS at fast speeds has been proposed as the trigger to use DP in flat terrain [16, 17]. However, this has been argued against by Ettema, Kveli [15], who found the same transitions at different speeds. If one transition occurred at critical speed, another one must have been at slower or faster speeds—suggesting that a unique critical speed does not exist.

Two main study design approaches have been used to try and disentangle the problem of technique transition triggers in XCS. One approach has involved prolonged conditions allowing athletes to choose or indicate the technique they prefer. Another approach has been to apply relatively rapid changing conditions over time and record the conditions at which athletes change technique spontaneously. This acute approach seems better for quantifying parameter values for technique transitions, but less suitable to address physiological optimisation issues.

Our research group tried to elucidate whether speed or incline was the predominant (if not single) triggering factor. Ettema, Kveli [15] concluded that incline seemed to be the predominant trigger. In that study, inclines and speeds (applied on a treadmill while roller skiing) were set to obtain combinations of equal external mechanical power. This was done to exclude power from the experiment as a potential third external trigger. In this respect, Pellegrini,

Zoppirolli [16] were not able to distinguish the effect of speed or incline from that of power. Of course, the evident limitation of [15] is that the role of power itself was not elucidated. Løkkeborg and Ettema [9] circumvented this problem by using an extended protocol that maintained either incline, speed, or power constant, by modifying two of the factors. The findings of that study only partly supported the idea that incline is the dominant factor. It was further concluded that power played a minor role in choice of technique. This conclusion is supported by the finding that both at extreme intensities during competitions (high external power) and during submaximal intensities (like in the aforementioned studies) the same techniques, DP, DK and DS, are chosen [19]. Still, in all these experiments changes in power were always associated with changes in incline and/or speed and so the isolated role of power was not clear.

Transition hysteresis, a difference in transition conditions between one direction compared to its reversed direction, in skiing only seems to occur when power is changing during a continuous protocol, not when kept constant while other parameters (incline, speed) change [9]. This is an indication that the role of power in influencing the choice of technique must not be excluded a priori, even if it seems to be of less importance than speed and incline [9]. To resolve this issue, we executed a protocol that changed power at fixed speeds and inclines.

Thus, the aim of this study was to examine the isolated role of power on the choice of technique during classic technique roller skiing, when power was changed at fixed speeds and inclines. This was achieved by applying different backwards pulling forces (approximately parallel to the treadmill surface) while the athlete roller skied at fixed speeds and inclines. The comparison of four such conditions (2 speeds x 2 inclines) allowed us to distinguish the specific effect of power from the speed/incline effects. Our overall thinking was that if athletes change techniques at the same power, despite differences in speed and incline, power may be regarded an independent trigger for transition. We hypothesized that at different fixed speed and incline combinations, technique transitions would occur at the same power.

## Methods

### Participants

Six male (24.1 ± 2.1 yrs. 77.6 ± 4.4 kg 180.5 ± 1.7 cm) and eight female (21.7 ± 1.7 yrs. 62.6 ± 7.1 kg 169.7 ± 3.6 cm) cross-country skiers participated in the study. All participants were members of the Norwegian University of Science and Technology student ski team, well-trained in XCS and familiar with roller-skiing on a treadmill. The study was approved by the Norwegian Social Science Data Service (NSD), ref.nr 512269, and the experiments were conducted in accordance with the Helsinki Declaration. The participants signed a written informed consent and were allowed to withdraw from the project at any time without providing a reason.

### Protocol

The tests were performed on a motorized treadmill (Forcelink Technology, Zwolle, The Netherlands) using a safety harness. All participants used the same pair of IDT roller-skis (IDT sports, Lena, Norway) equipped with wheels of rolling resistance category 2. A warm-up period assured that rolling friction coefficient was constant ($\mu \approx 0.018$) [9] during the experiments. The athletes could use their own poles or choose poles provided by the laboratory at length interval of 5 cm.

For each participant, the experiments were performed within one hour, including warm-up. During a ten-minute warm up, the skiers were acquainted with the treadmill and performed classic roller skiing at different inclines and speeds, assuring all techniques were used.

They were informed about the general design, i.e., during the exercise they were to roller ski (classic technique) freely, choosing a technique and execution as they saw fit while some backwards pulling resistive force was added or removed at certain time-intervals. All powers were estimated to be submaximal, although the highest ones might have exceeded some individuals' anaerobic thresholds. They were further informed that each bout (i.e., one speed-incline combination) was not expected to exceed seven minutes. The four combinations of two speeds (10 km h$^{-1}$ and 12 km h$^{-1}$) and two inclines (5% and 8%) were studied. These combinations were selected based on our previous studies [9, 15], assuring that they would be submaximal efforts, and the expectation that most athletes would not choose DS technique at the onset of the protocol because DS is usually chosen on steeper inclines.

A specially built pulley system was used to guide a rope between the athlete and a hanging weight. Using a broad belt, the rope was attached at the waist, just above the pelvic region, close to the assumed centre of mass location. The height of the pulley system was adjusted so that the pulling force was applied approximately parallel to the treadmill surface at both inclines. Minimal friction of the pulley wheel was assured by using a road-bicycle wheel for that purpose. The weights consisted of small bags filled with sand, and weighing 500 g, that could be manually added to or removed from the rope with ease. A 500g hanging weight corresponded to a ~13–16 W change in external mechanical power. Approximately every 20 seconds an additional weight was added to the rope until the athlete had adopted the DS technique. In such case, one more 500 g resistance mass was added at the same time interval to check if this DS technique was maintained. Occasionally, on the 5% incline, an athlete did not choose the DS technique even after adding considerable resistance. In such cases, the bout was terminated. Before including this outcome in further analysis, it was, after consultation with the athlete, affirmed that the athlete indeed would not have changed to that technique at higher resistance, and it was affirmed that the final obtained power exceeded that of transition to DS at other conditions (also see statistical analysis below).

## Recording equipment

A load cell (N-DTS-FS5, Noraxon USA Inc., Scottsdale, Arizona) connected in series in the rope-connection between pulley and athlete recorded the actual resistive force at 1500 Hz. Kinematic data were collected with Oqus 3D motion capture (Qualisys AB, Gothenburg, Sweden) using eight cameras at 200 Hz. Markers at the tip end of the poles and on the skis were used to identify the techniques using a custom algorithm [15] in Matlab (R2019b, Mathworks Inc., Natick, MA, USA).

The experiments were performed at the core facility NeXt Move, Norwegian University of Science and Technology (NTNU).

## Signal treatment and statistical analysis

Both resistive force and power were statistically analysed as potential triggers. While resistive force was recorded directly, the external mechanical power was estimated according to [20] and adding the power required to overcome the resistive force ($F_p$):

$$P = v \left( mg(sin\alpha + \mu \, cos\alpha) + F_p \right) \tag{1}$$

Where $m$ is effective body mass (including ski equipment), $v$ is treadmill belt speed ('velocity'), $g$ gravitational acceleration, $\alpha$ angle of incline, $\mu$ coefficient of rolling friction, $F_p$ the force on the pulley rope. Note that the different incline-speed combinations result in different resistive force-power combinations, allowing us to distinguish the role of power from the mechanism

that it was induced with (additional external resistive force). Even though our main aim was to investigate the role of power, it was also possible to elucidate the means by which power was altered, i.e., the resistive force. All athletes underwent the same resistive force increment despite considerable differences in body mass. Thus, there were considerable interindividual differences in total resistance, i.e., including gravity. Therefore, alongside additional resistive force, total resistive force (i.e., pulley force, gravitational component, and roller friction: $mg$ $(sin\alpha + \mu\, cos\alpha) + F_p$ was also investigated as a possible trigger. It should be noted that, in this study, only the mean external mechanical power is examined; any fluctuations in power due to the motion of limbs relative to the centre of mass, mechanical energy fluctuations of the body and associated losses were not considered. Because all classic XCS techniques include at least one propulsion and one glide phase during each stride, the additional resistive force fluctuated considerably during one stride cycle. Therefore, force and power time traces were smoothed using a moving average with a time window of 2 seconds. Force and power at the time of technique transition was averaged over a period of –1.5 to +1.5 seconds (i.e., period of about 1–2 whole stride cycles) around that time point. The main outcome of this procedure is shown in Fig 1.

The main statistical tests were comparisons between the external resistive forces and mechanical powers at the technique transitions across the two incline and two speed conditions. If force and/or power is to be regarded as an isolated trigger factor, the force and/or power at which transitions occur should stay constant across different incline-speed combinations. Thus, any difference in the external resistive force or mechanical power at the technique transitions between the four incline-speed conditions was regarded as evidence that the variable is not an isolated trigger factor. In some cases, an athlete did not choose all three techniques (i.e., they did not choose DP or DS) in all four conditions. In such a case, obviously a DP-DK or DK-DS transition was absent. Thus, even though the athlete exhibits behaviour (i.e., not changing technique) that differs from the behaviour in another condition, i.e., necessary and valuable information is available, it would initially appear as 'missing value' (no number was associated with such behaviour). To allow us to use such 'missing value' in the statistical analysis, it was filled in with a fictitious value, one force or power step higher than the maximal value or one step less than the minimal value observed for that athlete during all four incline-speed settings. A Bayesian 2x2 ANOVA for repeated measures (RM) was used to estimate means (± 95% credible intervals, CI). We report Bayes factors for inclusion ($BF_{in}$), quantifying the evidence (odds) for the model including incline or speed as predictors given the data (i.e., change from prior odds (0.5) to posterior odds [see e.g., 21–23]). We also report classical F- and p-values (2x2 RM ANOVA) as well as partial eta squared ($n^2$). The Bayesian approach quantifies evidence both in support of (the power and force at transition are equal in different conditions) and against the null-hypothesis (the transition power and—force differ between conditions). To avoid dichotomization of evidence and both type I and type II errors, we used a more 'holistic' approach when judging whether force and power at technique transition change or not [see e.g., 24]. Therefore, we focus both on $BF_{in}$, p-values but also on the actual differences and spread (95%CI), as well as data quality. For the Bayesian ANOVA, the prior for fixed effects (probability of incline or speed as predictors) was set at 0.5 [25]. For the main tests (effect of incline and speed on power and force at transition), the assumption of approximately normally distributed residuals was confirmed by checking Q-Q-plots visually.

To test for technique transition hysteresis, the external resistive forces at the technique transitions were compared between when the force was being incremented vs. decremented using Bayesian paired t-tests. $BF_{10}$, i.e., the relative likelihood of the alternative versus the null hypothesis, is reported with a Cauchy (r scale 0.707) prior [26]. RM ANOVA and paired t-tests

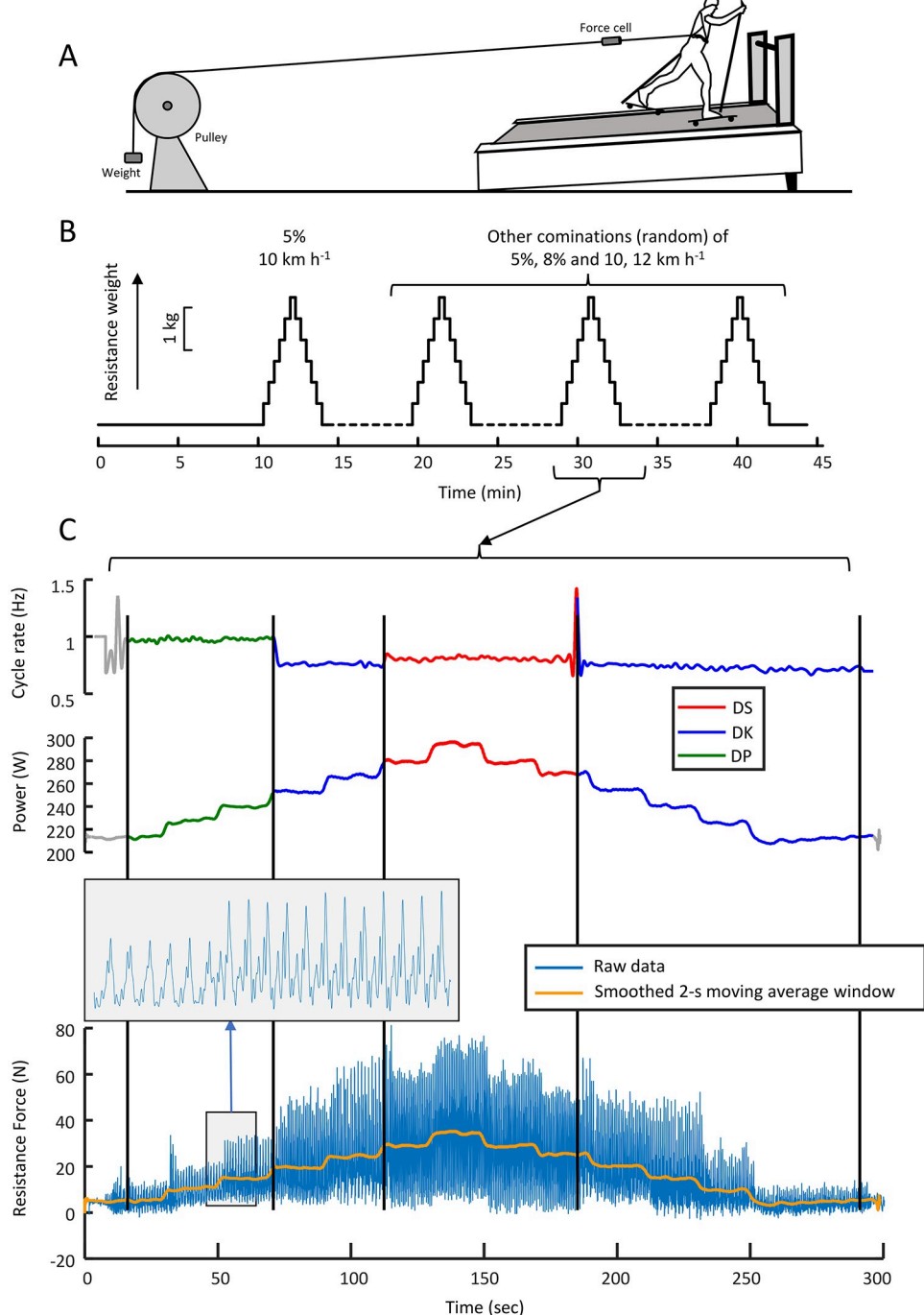

**Fig 1. Equipment setup, study design, and original time traces (cycle rate, power and additional resistive force) during one session.** A. Athlete on treadmill with hanging resistance weight. B. Timeline of protocol showing resistance weights. C. Cycle rate and power time trace (at v = 10 k h⁻¹, 8% incline). The techniques are indicated by colour. Bottom diagram shows the raw resistive force data (also in zoomed range) and smoothed over a 2 s window.

## Results

Fig 2 shows the distribution of the highest and lowest powers chosen in each technique. A clear and significant overlap of this power range exists among the techniques. That is, the

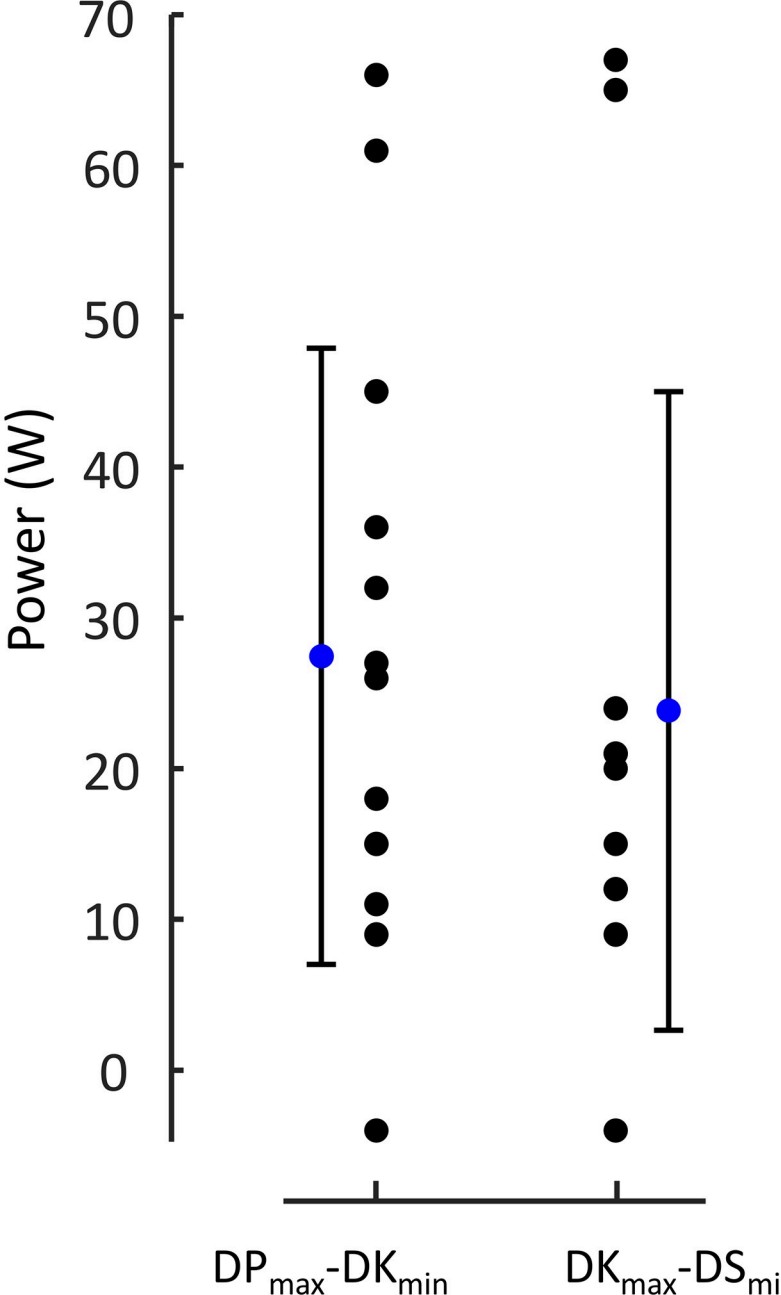

**Fig 2. Mean, SD and individual values of overlap of power distribution for the three techniques.** Differences between maximal and minimal power for 'adjacent' techniques are shown (irrespective of which session these were obtained).

maximal powers used in DP were higher than the lowest in DK (paired t-test n = 13, p<0.001) and the highest when DK was used was higher than the lowest power in DS (paired t-test n = 12, p = 0.001). No such overlap was seen between DP and DS, the maximal power in DP and minimal in DS appeared very similar (228 ± 55 W vs 223 ± 49 W; paired t-test n = 12, p = 0.438). Yet, despite the overlaps, the mean powers at which the techniques were used differed (One-way ANOVA repeated measures, n = 12, p<0.001, post-hoc p<0.01 for all comparisons). Fig 3 shows the relative time spent in the three techniques, per condition. By definition, the minimum and maximum values are 0 and 1, respectively, as indicated in the boxplot (Fig 3). The time using DP and DS were both affected by incline (n = 12, p<0.001, $\eta^2$>0.72, $BF_{in}$>182 for both). Time using DS increased with incline, time using DP decreased. Speed had no significant effect on time using the techniques, but evidence for not being affected is 'anecdotal' (n = 12, DP: p>0.470, $\eta^2$<0.04, $BF_{in}$ = ~0.4 for both). Similarly, time using DK was not significantly affected by incline (n = 13, p = 0.701, $\eta^2$ = 0.01, $BF_{in}$ = 0.439; speed: n = 13, p = 0.296, $\eta^2$ = 0.09, $BF_{in}$ = 0.462).

Not surprisingly, many athletes did not choose all technique transitions in all four conditions (11 for DK-DP and 8 for DK-DS; see also Fig 3, with maxima of 1 and minima of 0 for time spent in sub techniques). Furthermore, one athlete used DS in all conditions, and one never used DS, only DK and DP. The 2x2 ANOVA was applied to a subset of athletes who adopted sufficient transitions to fill a 2x2 matrix (DP-DK n = 10, DK-DS n = 11). Fig 4 shows, based on this subset, the individual and mean (±SD) values of power at which technique transitions occurred in the four incline-speed conditions. In Table 1, the statistical outcomes of the ANOVA are presented. The results are based on combined transition data of increasing and decreasing resistive force (DP-DK n = 10, DK-DS n = 11).

Additional resistive force and power at transition were affected by incline, but almost in an opposite fashion; at 8% incline, additional resistive force at technique transition tended to be lowest, while external power at techniques transition was highest. Power at technique transition was also affected by speed, which was not the case for additional resistive force, especially for the DP-DK transition. Total resistive force showed the same outcomes as power regarding incline effects. This was expected given Eq (1) but also infers that additional and total resistive force at transition are affected by incline in opposite ways. Speed effects on additional and total resistive force at transition are nearly identical.

The hysteresis outcome (mean per athlete over all four conditions) is presented in Fig 5. Both transitions showed positive hysteresis, i.e., the athletes retained a technique at greater resistive force (and thus external power—for hysteresis within one speed-incline condition, analysis of force and power lead to identical statistical results) when resistance was incremented versus decremented (paired t-test n = 12, DP-DK n = 13, $BF_{10}$ = 27.3, p = 0.007; DK-DS n = 12, $BF_{10}$ = 7.64, p = 0.002).

## Discussion

The aim of this study was to examine the independent role of mechanical power on technique transitions in classic style cross-country skiing. The findings confirm the general expectations regarding technique preference [e.g., 10]: DP was chosen at the lower power output range, DK at medium powers, and DS was chosen at the higher powers. However, the overlap of the ranges was substantial (Fig 2), confirming the overlap previously shown for speed [12]. This suggests that the factors that determine external power (i.e., incline, snow conditions, speed) are all important for the choice of technique [e.g., 15, 17]. This was also confirmed here (athletes tended to use the DS technique more at the steeper inclines, DP at the shallow inclines; Fig 3). Power at transition was about equally as much affected by speed and incline (although

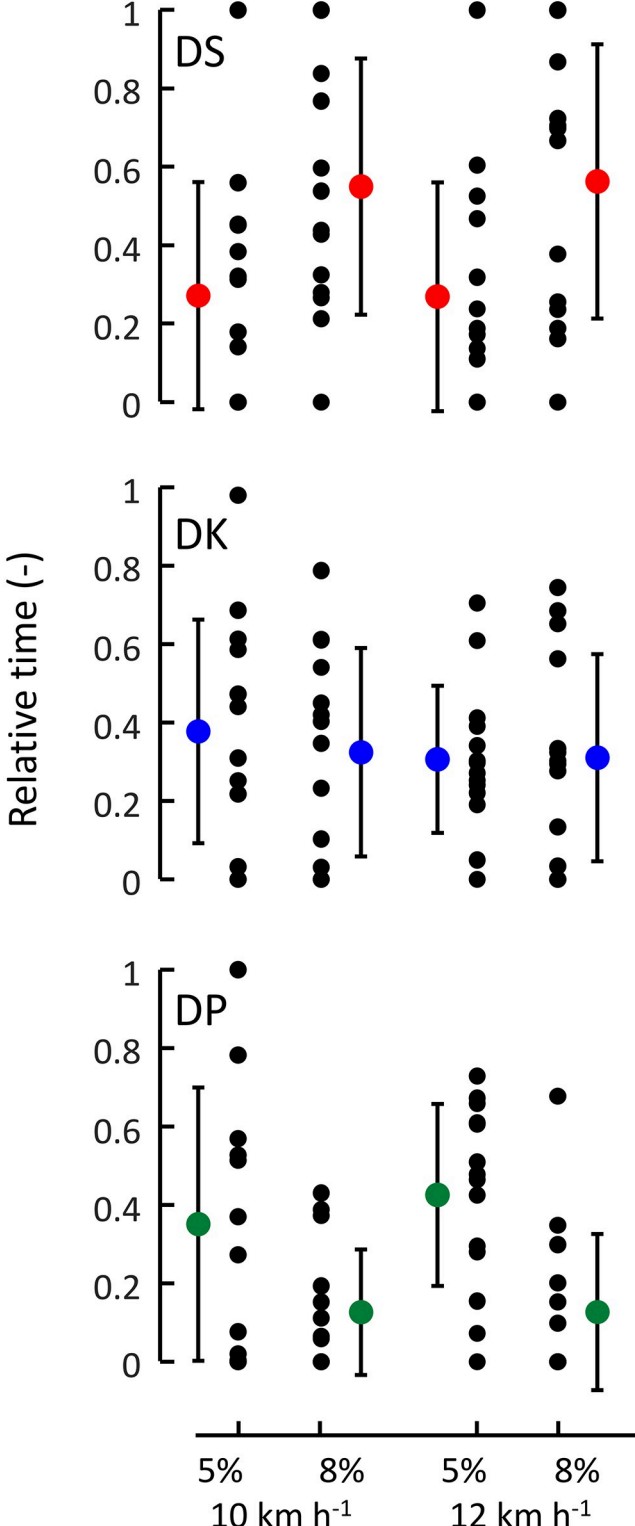

**Fig 3. Mean, SD and individual values of relative time that the different techiques were applied during the four protocols (2 speeds x 2 inclines).** Color signature for techniques is the same as in Fig 1. The individual values often range from 0 and 1.

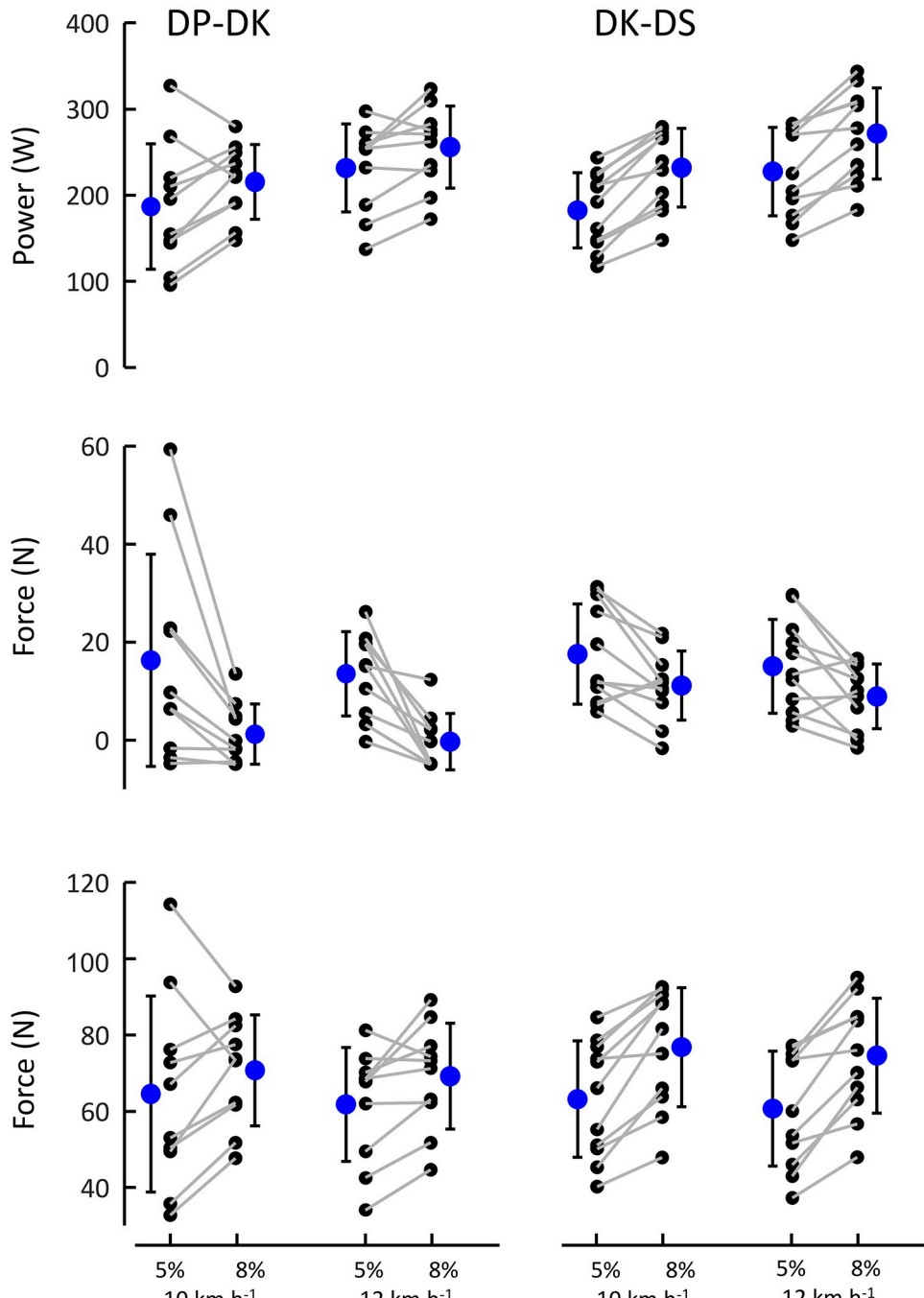

**Fig 4.** Mean, SD and individual values of (A) mechanical power, (B) additional resistive force and (C) total resistive force at technique transitions. Grey connecting lines indicate individual data from the same athlete. For reasons of clarity, these are only shown between different inclines at the same speed.

to a somewhat lesser extent by incline for the DK-DP transition; Table 1 and Fig 4). Both additional and total resistive force at transition were also strongly affected by incline but in opposite manners, while minimally affected by speed. The outcomes for total resistive force at transition are in accordance with Eq (1): at fixed speed, results are similar to power at

**Table 1. Statistical outcome of 2x2 RM ANOVA: Bayesian estimated mean (95% CI) of the differences (8%—5% and 12–10 k h$^{-1}$), BF$_{inclusion}$, classical F-values ($\eta^2$) and p-values.**

| Transition | Bayesian RM ANOVA | | Classic RM ANOVA | |
|---|---|---|---|---|
| | Incline | Speed | Incline | Speed |
| *External Power* | | | | |
| DP—DK | 22.8 (5.2–40.4) W 3.2 | 39.4 (20.4–57.8) W 37 | $F_{1,9}$ = 7.4 (.45) P = 0.024 | $F_{1,9}$ = 22.3 (.72) p<0.001 |
| DK—DS | 46 (36.2–55.4) W 4280 | 41.2 (31.6–50.8) W 13920 | $F_{1,10}$ = 75 (.88) p<0.001 | $F_{1,10}$ = 125 (.93) p<0.001 |
| *Additional Force* | | | | |
| DP—DK | -13.7 (-20.6 –-2.6) N 16.8 | -1.6 (-7.6–4.2) N 0.5 | $F_{1,9}$ = 16 (.64) p = 0.003 | $F_{1,9}$ = 0.4 (.05) P = 0.532 |
| DK—DS | -5.8 (-9.2 –-0.6) N 8.2 | -2.0 (-4.8 –-0.8) N 0.8 | $F_{1,10}$ = 11 (.53) p = 0.007 | $F_{1,10}$ = 4.8 (.33) p = 0.052 |
| *Total Force* | | | | |
| DP—DK | 4.6 (-1.8–11.2) N 1.5 | -1.4 (-7.4–4.2) N 0.5 | $F_{1,9}$ = 4.2 (.32) p = 0.070 | $F_{1,9}$ = 0.4 (.05) p = 0.532 |
| DK—DS | 13 (8.4–17) N 1258 | -1.6 (-4.6–0.8) N 0.9 | $F_{1,10}$ = 54 (.84) p<0.001 | $F_{1,10}$ = 4.8 (.33) p = 0.052 |

transition; at fixed inclines, they are almost identical with additional (pulley) resistive force. This last similarity indicates that body mass differences—and thus differences in gravitational and roller friction resistance—between athletes had little impact on the outcomes. This occurred, even though gravitational and roller friction resistance (at transition) comprised at least about half of total resistive force (Fig 4B vs 4C).

## Potential triggers for technique transition

Regarding potential external triggers for technique transition, it should be noted that we tested the null hypotheses that technique transitions would occur at similar resistive forces and power outputs regardless of incline or speed. The challenge of this approach was explained earlier [9] and the interpretation of the classical statistical outcome (p-values) is based on the process of elimination: any factor (resistive force or power output at transitions) showing significant differences between conditions is not a trigger, but the other factors might be. In this regard, Bayes factors are helpful as they provide evidence, both for and against, the null hypothesis, i.e., equality of power and force at transition, independent of condition. Keeping this in mind, our interpretation is as follows. 1: In conditions where only speed was changed, resistive force (both additional and total) may be a trigger, but power is not. Comparing transitions at the same incline but different speeds, athletes tend to shift technique at same resistance, not the same power (Fig 4, Table 1). Referring to the Bayes factor values for the speed effect on force at transition, it should be noted that only anecdotal evidence is available supporting the null-hypothesis (identical force at transition) compared to the alternative (Table 1: BF$_{in}$ = 0.5–0.9 –a value of at most ~0.2 would suggest otherwise). In other words, we cannot conclude that resistive force definitively has a trigger function, but the possibility for such is present. 2: In conditions where only incline was changed, neither force nor power is a clear candidate for this trigger function (possibly with the exception of total resistive force in DP-DK transition) (Fig 4, Table 1). These findings seem to agree with earlier studies: Both Ettema, Kveli [15] and Løkkeborg and Ettema [9] indicated that incline seemed to play a bigger role for technique transitions than did speed or power. Yet, as mentioned, the design of these studies did not allow any conclusion concerning power as such a factor in isolation, i.e., obtained independently of incline and speed.

## The role of external power

The current results (Fig 4, Table 1) confirm that external mechanical power generally cannot be regarded as a trigger for technique transitions. External power is calculated from a

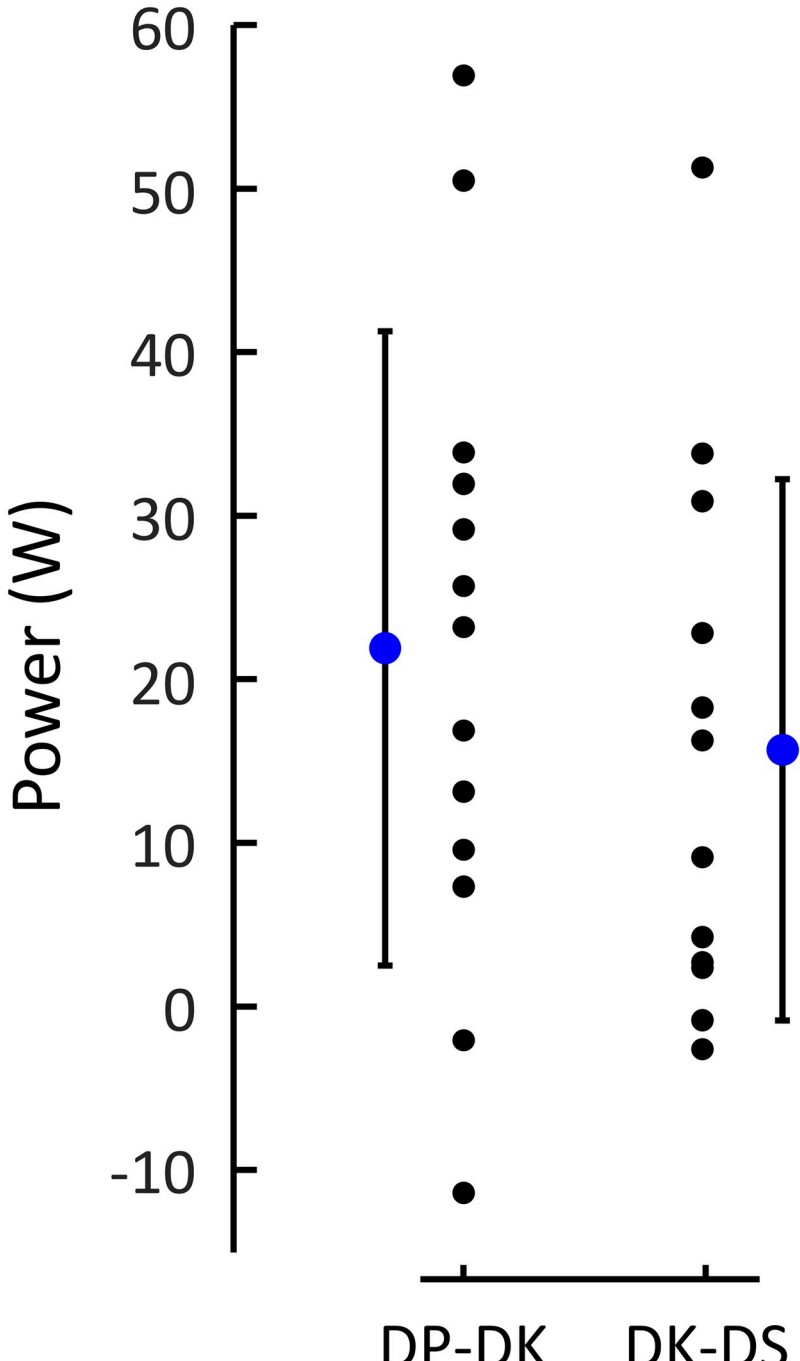

**Fig 5. Mean, SD and individual values of mechanical power difference for technique transitions when increasing and decreasing resistive force.** The data are the means over the four protocol sessions per athlete.

combination of speed, incline and resistive force. In contrast to our previous studies [9, 15], here power was manipulated independent of speed and incline. Still, the most likely conclusion supports our previous findings: rather than mechanical power output itself, one or more of the underlying factors (e.g., external resistance) may be a trigger for technique transitions. The

marginal role of power in the choice of technique is substantiated by the fact that in this and previous studies, all techniques were chosen even though intensity was only moderate. Further, in XCS competitions, when intensity is clearly very high, all three techniques are chosen under different conditions (terrain, snow friction, wind etc.) [19, 27].

The finding that external power itself unlikely functions as a trigger for technique transitions is comparable to those found in studies on the walk-run transitions. Even though people tend to use the gait (walking vs. running) that is energetically most economical, energetics are not a direct trigger to transition between gaits. The change in movement economy (i.e., $\dot{V}O_2$) changes too slowly to be considered a trigger for an abrupt gait transition [e.g., 2, 4, 6, 28]. As in walk-run transitions, other variables that do change more abruptly (i.e., muscular processes —the origin of $\dot{V}O_2$ changes) are more likely triggers, though not easily measured.

## Potential mechanisms

The role of resistance force, used as mechanism to change power independent of speed or incline, was investigated as secondary aim of study because it potentially could assist elucidating mechanisms that may be responsible for choice of technique. In contrast to power, almost identical resistive forces were found on average at DK-DP and DK-DS transitions between speeds at the same incline, but there were considerable interindividual differences. The power jumps per weight added differed by only 3 W at the two treadmill speeds. Thus, this clear discrepancy between power and force concerning the effect on behaviour can hardly be explained by this small protocol anomaly. Therefore, external resistive force may play a trigger role on a fixed incline. As mentioned, there is no evidence for either outcome ('force definitively has no trigger role' vs. 'force clearly has a trigger role'). Yet, the difference between power and additional resistive force in this respect is distinct. These findings strengthen the notion of the strong trigger function of incline—and possibly the associated mechanism of using the body's mechanical energy in 'falling' on the poles that is more effective at shallower inclines [see 9, 18]. Beyond that, a direct trigger role of resistive force (when incline is not in play) does not necessarily indicate that the sense of resistance (in this case around the pelvic region) is the intrinsic trigger. When maintaining speed, as was done in all experiments, total resistive force is uniquely related to propulsive force. The sense of having to apply increasing propulsive force is the corresponding internal trigger candidate.

Moreover, we cannot induce any general principle about resistive force, i.e., independent of the way additional resistive force was applied (here close to centre of mass). In this respect, the current study cannot elucidate if additional or total resistive force is the key for such 'sense of effort'. Still, the fact that additional force and total force (and power) at transition show opposite effects of incline (Fig 4) suggests that the way resistance is applied is plays a role in this 'sense of effort' leading to the choice of technique. However, as mentioned, the current study was not designed to investigate this detail explicitly.

The current findings support the notion presented by Pellegrini, Zoppirolli [16], i.e., maintaining (pole) forces—which are directly linked to resistance forces in DP as applied in this study —below a certain threshold serves as a trigger. In the present study, athletes changed from DP, via DK, to DS with increasing demand for propulsion, independent of incline. In previous studies [9, 15], athletes did the same, but the increased (propulsive force) demand was induced by increasing incline. In DP, propulsion through the skis is not possible. The only way for power from the legs to contribute to propulsion through the poles is by the utilization of body's mechanical energy, i.e., "falling on the poles". Therefore, in those studies, we opted for the notion that this mechanism becomes less effective at steeper inclines [17]. This leads to the choice of DS because the legs can generate considerably more power directly through ski push-offs [17].

The current findings suggest that at least one other mechanism contributes to triggering the transition to DS on steep inclines. We cannot decipher from our data if indeed a maximal threshold for poling force plays a key role [16]. However, the notion that poling force acts as a trigger seems plausible. Even though the lower extremities can contribute profoundly to power production in DP [18, 29–31], all propulsive force ultimately is applied via the poles, placing considerable demands on the upper extremities. At the higher resistances induced by incline, upper extremity muscle activity (and the time of force generation) may become too great [18] and by transitioning to DS, the propulsive force can be distributed between both poles (~arms) and skis (~legs). Thus, while we previously found evidence against a pole force trigger [17], the current study—using a different protocol paradigm—to some extend supports the pole force trigger hypothesis. This idea is supported by Andersson, Pellegrini [32] who found that pole forces in DS increased with increasing speed, likely related to a reduced propulsion time [32] (and may explain why DS usually is not chosen at high speeds).

## Hysteresis

Based on our previous studies [9, 15], we expected hysteresis to occur because power changed during the protocols. This expectation was confirmed, and once more the hysteresis effect was rather small but relatively consistent. Our present data suggest hysteresis not only occurs when incline is changed but applies to a wider 'terrain' of conditions. Like in walk-run gait transitions, hysteresis is an inherent and complicating aspect of technique transitions in XCS.

## Atypical behaviour

Curiously, one of the athletes never used—and another one only used—DS. No obvious difference in demographics or experience was noted for these athletes. Obviously, they were not included in the statistical analysis for transitions they never made. This also applies for a few other athletes who used only one technique in some of the conditions. This is the reason for the discrepancy between number of athletes that participated and the n-value for the analysis (14 vs 11 and 10), which was not due to technical errors. The behaviour of these athletes should not be disregarded in the interpretation of the findings. It rather highlights the complexity of the mechanism behind technique transitions. Likely, for these athletes, the range of the speed-incline-resistance conditions was outside of their individual range for one or two of the techniques.

## Methodological considerations

Like in our previous studies, we made stepwise changes in one of the external conditions, in this case external resistive force. It is not clear if a continuous change would have affected the outcome of study. However, as indicated previously [9], we have no evidence of that being the case. The identification of the exact resistive force and power at spontaneous transitions would have been more precise, but not essentially different. The outcome of this study confirms that the current sensitivity was sufficient for detecting the role of power. A study with higher force change resolution, i.e., smaller weight increments, may elucidate if indeed force is (close) to identical for DK-DP transitions at different speeds, or whether small force differences—undetectable in the present study—occur.

The choice of speeds and inclines was based on previous studies [9, 15], but resulted in larger power differences between the two inclines than the two speeds (at the same resistance force). This raises the question if speed differences would have been somewhat larger, would the statistical outcome for speed have resembled more closely that for incline (Table 1) and disqualified resistance force as trigger for techniques transition as well? One may argue that this

would not have an influence since power itself is not a factor of importance regarding triggering technique transition. In any case, the current findings are not conclusive about the role of resistance force.

The continuous recording of resistive force (Fig 1) shows large force fluctuations (and thus power fluctuations). On might argue that these large fluctuations have affected the outcome of this study. However, the way resistance was applied may only have amplified such fluctuations, not caused them. Power fluctuations are inherent to the XC skiing techniques. For example, in this study, the body's speed changes during one cycle amounted around 1 m s$^{-1}$, which is associated with about 100 W power fluctuations (including potential and kinetic). The total power fluctuations registered were up to about 200 W, caused by fluctuations in additional resistive force (from the pulley). In other words, regardless the locomotion form, power fluctuation during one propulsion cycle is an inherent aspect of locomotion, and definitely in classic cross-country skiing [17].

Our sample size was relatively small (N = 14). Thus, some robustness assessment was warranted. Therefore, we used JASPs robustness check software to see how statistical outcomes depended on prior width and N. Although statistical outcomes are vulnerable at low N, for no variable was our conclusion affected in any way when reducing N down to 7 or 8.

## Concluding remarks

In all, considering the previous studies and the current one, the notion that a single factor may act as trigger for transition seems naïve. However, based on the present study, we can now reject external mechanical power demand as an important trigger for cross-country skiing technique transitions. The 'sensation' of a technique becoming unsuitable for the external conditions at hand, likely has a multifactorial origin, including local muscle forces, (visual [33]) perception of speed and the gravitational field [see also 16]. In a similar way, walk-run transition speed is affected by incline [2], indicating that speed itself (or any directly associated factors) cannot be the sole external factor triggering gait transitions.

## Supporting information

**S1 Data.**
(XLSX)

## Author Contributions

**Conceptualization:** Gertjan Ettema, Knut Skovereng, Tobias Ritman, Jørgen Danielsen.

**Data curation:** Gertjan Ettema.

**Formal analysis:** Gertjan Ettema, Jørgen Danielsen.

**Investigation:** Gertjan Ettema, Knut Skovereng, Tobias Ritman, Jørgen Danielsen.

**Methodology:** Gertjan Ettema, Knut Skovereng, Tobias Ritman, Jørgen Danielsen.

**Project administration:** Gertjan Ettema.

**Software:** Gertjan Ettema.

**Supervision:** Gertjan Ettema, Knut Skovereng, Jørgen Danielsen.

**Validation:** Gertjan Ettema, Knut Skovereng, Jørgen Danielsen.

**Visualization:** Gertjan Ettema, Tobias Ritman.

**Writing – original draft:** Gertjan Ettema, Knut Skovereng, Tobias Ritman.

**Writing – review & editing:** Gertjan Ettema, Knut Skovereng, Jørgen Danielsen.

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
