## [Decision Letter · Decision Letter 0]

26 Sep 2022

PONE-D-22-10768The role of work rate on the choice of technique in classic roller skiingPLOS ONE

Dear Dr. Ettema,

Thank you for submitting your manuscript to PLOS ONE. After careful consideration, we feel that it has merit but does not fully meet PLOS ONE’s publication criteria as it currently stands. Therefore, we invite you to submit a revised version of the manuscript that addresses the points raised during the review process.

Two out of three expert reviewers recommended major revision, whereas the third recommended rejection. Despite this, I believe MS has some merits. However, authors need to address all reviewers' issues point-to-point (in particular Reviewer 2's).

We look forward to receiving your revised manuscript.

Kind regards,

Luca Paolo Ardigò, Ph.D.

Academic Editor

PLOS ONE

2.Please include captions for your Supporting Information files at the end of your manuscript, and update any in-text citations to match accordingly. Please see our Supporting Information guidelines for more information: http://journals.plos.org/plosone/s/supporting-information.

Additional Editor Comments:

Two out of three expert reviewers recommended major revision, whereas the third recommended rejection. Despite this, I believe MS has some merits. However, authors need to address all reviewers' issues point-to-point (in particular Reviewer 2's).

Reviewers' comments:

Reviewer's Responses to Questions

**Comments to the Author**

1. Is the manuscript technically sound, and do the data support the conclusions?

Reviewer #1: Yes

Reviewer #2: Partly

Reviewer #3: Partly

2. Has the statistical analysis been performed appropriately and rigorously? 

Reviewer #1: I Don't Know

Reviewer #2: No

Reviewer #3: Yes

3. Have the authors made all data underlying the findings in their manuscript fully available?

Reviewer #1: Yes

Reviewer #2: Yes

Reviewer #3: Yes

4. Is the manuscript presented in an intelligible fashion and written in standard English?

Reviewer #1: Yes

Reviewer #2: No

Reviewer #3: Yes

5. Review Comments to the Author

Reviewer #1: Overall Comments

I find this manuscript interesting both as a scientist who studies both bipedal human locomotion and xc skiing. The approach and findings have implications for other scientists who study only bipedal locomotion and those who are more sports oriented. Thus, the topic seems appropriate for a general audience journal like PLOS ONE. The study design is creative and well-executed. The presentation (writing and graphs) need considerable improvements! But, that’s OK because the content is interesting and important.

I have zero expertise about Bayes statistics, maybe a statistician should take a look.

General Comments:

1. would like to see a simple diagram of the experimental set-up, just to clarify that the rope pulled backwards parallel to the treadmill surface.

2. I fully understand that power is the rate of performing mechanical work. Currently, there is a mix of “work rate” and “power”. The authors should use ONE term or a shortened version throughout the paper and not mix the two. I think “external mechanical power” is the most accurate and best term to use in this paper which could and should be defined and then a shortened version could be used, e.g. “We measured average external mechanical power as the product of the summed (or total) resistive forces (gravity, rolling friction and the applied load) and the treadmill belt velocity. We use the shorter term “power” for simplicity.” Of course, the authors may prefer to use mechanical work rate throughout. In my specific comments, I have sometimes written power and sometimes kept the authors’ use of work rate just to make it clear what other changes I am actually suggesting. Overall, please pick power or work rate and stick with it.

3. The authors should consistently use EITHER “additional resistive FORCE” or “additional resistive LOAD” but not both. I would prefer force.

4. It would be helpful if the authors noted that their method does not take into account within stride fluctuations in the kinetic energy of the COM. Perhaps they could also estimate how large/small that quantity typically is.

5. I suggest that the flow of the Introduction could be improved, especially for a non-specialist readership. I would suggest that the order should begin with the familiar (human walk-run transition), touch on quadrupedal gait transitions and then onto the less familiar classic Nordic skiing sub-technique transitions.

6. Again, try to use just one term “trigger” rather than sometimes trigger and sometimes “controlling factor/parameter etc.” I prefer trigger.

FIGURES

Figure 1 needs a label for horizontal time axis units, I believe they are seconds.

Figure 2 Colors {Grey/Green for DP, Green/Purple for DPK and Purple/Grey for DS} are which are not the same as Figure 1 {green (DP), blue (DPK), and red (DS)}. Please make the colors more consistent. What is meant by grey?

Figure 3 & 4, then use green and blue but to indicate incline!? Either use different colors for incline or just add 5% and 8% text inside or near the boxes. Further, speed (velocity?) is now in m/sec but in Methods section etc. 10 & 12 km/hr. Be consistent!

Figure 5 uses grey but not with the same meaning as in Figure 2.

Table 1 caption: outcomes (plural).

Minor Stylistic Comments

TITLE:

PLOS ONE is a non-specialist journal. I am an avid Nordic skier and biomechanist but probably, more than 99% of the international readers have never roller skied or even heard of roller skis. I’d bet that with the current title, most PLOS ONE readers would guess that your research is about in-line roller skating. Probably <10% have ever enjoyed nordic skiing. But, many have seen/watched nordic skiing on television once every 4 years but most know it as “cross-country skiing”.

I encourage the authors to just use “classic cross-country skiing” in the title.

I understand that in the sport-specific Nordic skiing literature, the terminology is usually two “techniques” (classic and freestyle/skating) and then “sub-techniques” within each technique. However, in the title, I like how the authors just use “technique” because “sub-technique” would require an explanation to a general reader. As I discuss below, the authors should decide if they want to use work rate or work-rate or power or external mechanical power etc. etc. and then use that term in the title. I also suggest that the authors simplify the body of the paper by just using “technique” throughout.

ABSTRACT:

Here and throughout the manuscript: decide if you want to use “velocity” (v) or “speed” and then use it consistently, don’t switch back and forth.

Line 13 Add comma after cross-country skiing,

L13 suggestion: …use different techniques akin to locomotor gaits such as walking and running. (I don’t think the “gear” analogy is very effective on its own and requires an explanation).

L14 what do you mean by “demand”? why not just say “techniques generally depend on speed and incline.”

L15 roles (plural)

L15 “… roles of speed, incline and mechanical power demand in triggering technique transitions.”

L16 cut sentence beginning: However…

L17 change this sentence to “However, it is still not known if mechanical power demand, as an isolated factor, has any role on the choice of technique.

L19 …choice of technique during classic cross-country roller skiing by changing mechanical power demand at fixed speeds and inclines.”

L21 … at the four combinations of two speeds (10 and 12 km/hr) and two inclines (5 and 8%) while additional resistive forces were applied via a weight-pulley system.

L23 Athletes were free to choose between three techniques (double poling, double poling with kick….

L26 Obviously power is affected by both incline and speed, but the focus of this paper is the trigger(s) for technique transition. Thus, I suggest the overall findings be summarized as: “At a given incline, technique transitions occurred at similar additional resistive force magnitudes at the two speeds but on the steeper incline, the transitions occurred at smaller resistive forces. Transitions were not triggered at similar mechanical power demands across the different incline/speed/resistive force conditions. This suggests that mechanical power itself is not a key technique transition trigger.”

L30 “but only when incline is not involved” is very confusing.

L31 “…suggest that technique transitions in classic cross-country skiing are not triggered by a single factor.”

INTRODUCTION

L36 These phenomena are well-studied

L39 Yet, while minimizing metabolic cost…

L42 …challenge exists for understanding the transitions…

L42 I suggest that you don’t worry about skating, and cut the gear analogy. just write: In classic-style XCS, skiers make transitions between three main techniques: double poling (DP), ….

L46 cut “the heaviest gear”

L46 with propulsion generated…

L47 DP is usually chosen for flat/low angle terrain when speed is fast.

L48 cut “lightest gear”

L49 …leg movements, with both poles and skis transmitting propulsive forces.

L50 DS is usually chosen on uphill terrain when speeds are slow.

L51 cut gear

L51 similar to DP but with the addition of one ski thrust per poling cycle.

L53 more influential parameter (9,10).

L54 …are tightly coupled. Average external mechanical power is equal to the product of velocity and the sum of the resistive forces (i.e. gravity, friction and aerodynamic drag forces). Thus, even though techniques chosen are generally linked to incline, it is not apparent if one factor (incline, speed, or power) actually triggers the spontaneous transitions between techniques.

L58 cut sentence “Even less so…”

L60 move this walk-run sentence earlier in the Introduction, following my overall ordering suggestion.

L65 on steep uphill terrain and at slow speeds.

L65 This idea was challenged by Dahl & Sandbakk (12) who suggested that on steep uphills, the mechanism….

L67 specify: utilization the body’s gravitational potential energy

L70 on steep inclines

L72 at fast speeds

L73 on flat terrain (11,12). But, Ettema & Kveli (9) disagree and suggest that…. briefly explain their argument here.

L75 cut first sentence and replace with the following suggestion.

L76 Two main study design approaches have been used to try and disentangle the problem of technique transition triggers in XCS.

L78 …different but prolonged conditions allowing athletes to choose or indicate the technique they prefer. Another approach has been…

L81 This acute approach seems better for quantifying…

L84 our research group tried to elucidate…

L85 triggering factor

L86 predominant trigger.

L90 the evident limitation of Ettema & Kveli is that the role…

L91 provide names for (10), e.g. Lokkeberg & Ettema circumvented this…

L91 by using an extended protocol…

L93 only partly supported the idea that incline is the dominant factor.

L93 Lokkeberg & Ettema concluded that work rate alone played…

L94-97 I don’t understand this sentence/study and can’t help you to make it more clear.

L98 start a new paragraph for hysteresis and first define what it means.

L104 …protocol that changed work rate at fixed speeds and inclines.

L108 work rate was changed at fixed speeds and inclines.

L108 This was achieved by applying different backwards pulling forces (parallel to the treadmill surface) while the athlete roller skied at fixed speeds and inclines.

L110 The comparison of four such conditions (2 speeds x 2 inclines) allowed us to distinguish the specific effect of work rate from the speed/incline effects.

L112 Our overall thinking was that if athletes transition between techniques at the same….

L113 then, restate this thinking as a Hypothesis. We hypothesized that at different fixed speed and incline combinations, technique transitions would occur at the same work rate.

METHODS

L119 it sounds like some subjects were dedicated skiers whereas others might be footballers, gymnasts, swimmers???

L127 warm-up is hyphenated

L132 some backwards pulling resistance force was added…

L133 ones might have exceeded some individuals’ anaerobic thresholds. Subjects’ anaerobic thresholds were not measured...

L137 The four combinations…

L138 were studied. (note: don’t overuse “applied”. I suggest: The resistive force was “applied”, the technique were “chosen” and the conditions were “studied”.

L140 would not choose DS technique at the onset of the protocol because DS is usually chosen on steeper inclines.

L142 between the athlete and a hanging weight. Using a broad belt…

L145 specify that the height of the pulley was adjusted so that the pulling force was applied parallel to the treadmill surface. (p.s. good idea to use a bicycle wheel! Very low frictional torque at the rim.)

L145 The weights consisted of small bags…

L148 separate sentence: A 500g hanging weight corresponded to a ~13-16 W change in external mechanical power.

L149 at the same time

L150 if this DS technique was maintained

L150 Occasionally, on the 5% incline, an athlete did not choose the DS technique even after adding considerable resistance. In such cases, the bout was terminated.

L161 …own poles or choose poles provided … intervals of 5 cm.

L173 I suggest just using the one term “trigger” and not add another term for the same thing (control parameter)

L174 Here is a good place to define formally either “external mechanical power (or work rate)”

L174 it is also a good place to mention that other work is done (internal limb work, COM KE fluctuation work)

L177 probably should use “velocity” throughout the paper if you want to use v (which is fine). Should also specify that this v is the treadmill belt velocity.

L178 rolling friction

L180 allowing us to distinguish (and cut: as such)

L181 external resistive force, or external impeding force or whatever you decide to use consistently

L181 I’m confused here. DS involves two propulsion and glide phases per stride cycle. I suppose you could say “Because all classic xcs techniques include at least one propulsion and one glide phase per stride cycle, the external resistive force fluctuated …

L193 The main statistical tests were comparisons between the external resistance forces and mechanical powers at the technique transitions across the two incline and two velocity conditions.

L196 any difference in the external resistive force at the technique transitions or mechanical power at the technique transitions between the four incline-speed conditions is regarded as evidence that the variable is not an isolated trigger factor. In some cases, an athlete did not choose all three techniques (i.e. they did not choose DP or DS) in all four conditions.

L199 I really do not understand the sentence beginning: In such cases…

L205 I agree you are testing a null hypothesis, but you should have stated the Hypotheses way back in the Introduction.

L208 To test for a technique transition hysteresis, the external resistive forces at the technique transitions were compared between when the force was being incremented vs. decremented.

L209 no need to (Note...) parentheses.

RESULTS

L216 …powers for which each technique was preferred.

L222 powers at transition differed

L227 Time using DS … time using DP decreased

L228 Speed had no effect (on what?)

L229 time using DK was not significantly affected by incline: …0.308 or velocity: p=0.296…

L242 specify how many athletes

L242 X athletes did not choose to transition between all three techniques in all four conditions.

L248 outcomes of the ANOVA are

L249 Results section should all be in past tense: were affected

L249 explain what you mean by reversed fashion

L250 Power at technique transitions were also affected by velocity which was not the case or resistive external force…

L261 explain what a positive hysteresis means in words, i.e. something like: subjects retained a technique at greater forces when resistance was being incremented vs. decremented.

DISCUSSION

I usually write the Discussion in past tense while reviewing/summarizing the results.

L269 the independent role of mechanical power on technique transitions in classic cross country skiing.

L271 The findings confirm the general expectations regarding technique preference: DP was chosen at the lower power output range, DK at medium powers and DS was chosen at higher powers.

L273 ranges was

L274 mixing your terms here: load = ? terrain = ? I don’t understand this sentence.

L275 Athletes tend to choose the DS technique more on steeper inclines and DP on shallower inclines.

L279 Both power at transition and resistance force at transition were strongly affected…

L281 cut (control parameter)

L282 should be noted that we tested the null hypotheses that technique transitions would occur at similar resistive forces and power outputs regardless of incline or velocity. The challenge of this approach…

L285 any factor (resistive force or power output at transitions) showing significant differences between conditions is not a trigger, but the other factor might be.

L289 only velocity was

Page 14-15 -16 is a VERY LONG paragraph! Break it into several paragraphs each with a topic sentence.

L297 studies. Both Ettema & Kveli….

L300 This is the MAIN FINDING but it is buried in the middle of this gigantic paragraph. Try to give it more attention, probably as the topic sentence of a shorter paragraph.

L302 cut: “In the current study”

L304 (9,10), here mechanical power was manipulated independent of velocity and incline.

L304 Still, the most likely conclusion supports our previous findings: rather than mechanical power output itself, one or more of the underlying factors (e.g. external resistance) may be a trigger for technique transitions.

L308 were chosen despite the fact that aerobic intensity was only moderate. Further, in xcs competitions, when aerobic intensity is clearly very high, all three techniques are chosen under different conditions (terrain, snow friction, wind etc.).

L311+ Even though people tend to use the gait (walking vs. running) that is energetically most economical, energetics are not a direct trigger to transition between gaits. The change in movement economy (i.e. VO2) changes too slowly to be considered a trigger for an abrupt gait transition (see 16). Other variables that do change more abruptly (i.e. muscular factors) are more likely triggers.

L317 Again, this is a very long paragraph. I suggest breaking into multiple paras

L318 transitions between speeds at the same incline, there were considerable interindividual differences.

L319 differed by only

L320 at the two treadmill velocities.

L321 Therefore, external resistance force may play a trigger role on a fixed incline.

L324 …trigger role’). Yet, the difference…

L327 effective at shallower inclines

L329 is the intrinsic trigger.

L332 force is the corresponding internal trigger candidate.

L336 athletes changed

L337 to DS with increasing demand

L339 cut (and as mentioned above)

L340 …mechanical energy for propulsion via the legs becomes less effective...

L342 This leads to the choice of DS because the legs can generate considerably more power…

L344 other mechanism contributes to triggering the transition to DS on steep inclines. We cannot..

L346 key role (11). Cut rest of sentence

L348 extremities can contribute

L351 …and by transitioning to DS, the propulsive force can be distributed between both poles (arms) and skis (legs).

L352 Thus, while we previously found evidence against a pole force trigger (12),…

L356 likely has

L359 gait transitions.

L360 Curiously, one of the athletes…

L367 was outside of their individual range….

L367 was there anything notable about these athletes in terms of skiing experience/expertise?

L375 …conditions. Like in the walk-run gait transition, hysteresis is an inherent

L381 external resistance force

L383 cut Rather

L384 more precise, but not essentially different. The outcome of this study….

L390 plural: fluctuations

L393 “not generated” is confusing.

L396 I suggest a final summary paragraph.

Reviewer #2: Overall Comments

I believe that the authors’ research focus and the data originality are within the PLoS policy. The introduction section was well written and logical study necessity and hypothesis can be found except for some minor mistakes (P3 L50 … choice ‘of’ and … and P3 L66 … still very much… etc.). English and/or sentence editing is highly necessary even if you are native English speakers. However, probably due to a lack of appropriate number of participant, statistics and result sections are confusing.

Major comments

1 P3 L43. You need to present a sample movie to explain DP, DK, and DS.

2 P5 L123-L124. You should show the production name of the used treadmill and classic roller skiing at the first appearance, but not in P6 L151 and L153.

3 P5 L122-L135. You showed a typical example in Fig. 1, but explanations of a series of the protocols should be presented as a figure before presenting the current Fig. 1.

4. I seriously confused why the authors mixed to use Bayesian analysis, box plots, and 2×2 repeated measures of ANOVA without checking data normality for this small sample size. I hope you to analyze with usual statistics in biomechanics or sports physiology area. The result section and Figure presentations should be conducted from the scratch. Bayesian approach has been used for estimation of available parameters, future prediction of big data, choice of the mathematical model, and its statistical analysis. Even if your statistics is correctly used, more participants could be necessary.

4-1. The authors need to justify the estimation of the original sample size. In my calculation (effect size = 0.25, alpha level = 0.05, statistical power = 0.8, if correlations among repeated measures are high enough = 0.7-0.8), you need to test at least 16-18 participants.

4-2. The effect size (η2 values for ANOVA and Cohen’s d values for t-test) and degree of freedom need to be provided in this small sample size. Did you confirm the data normality before performing ANOVA? How were the F values? T-test (as a post hoc test in this study) also requires data normality. Note that the effect size and p-values are dependent on the degree of freedom.

4-3. Even though you tested small number of participants, you used 2-way repeated ANOVA before checking the data normality. In addition, why did you combine Bayesian approach and ANOVA?

4-4. There is a serious claim for presenting box plots in Figs. 3-5. DO NOT use the box plot in your case because it sometimes provides the same plot even if the data distribution is completely different. Herein, the problem came from the ref. #15. The #15 seemed to aim for a large sample size (200-5000) for PSYCHOLOGISTS. You used biomechanical technique, so I ask you whether it is appropriate for your study design (I do not think it is appropriate). Usual bar presentations are necessary instead of box plots.

5. P10 L260. In your statements in the introduction section, the aim of this study was to examine the role of “incline and speed”, but not “role of work rate”, wasn’t it? Check in detail.

6. English editing is highly necessary in the discussion section. It sometimes lacks subject, and too many spoken English apparently existed (e.g., P10 L275 … is likely not …; P10 L280 and L281 at ″the″ same…; P10 L287 a bigger role… should be rephrased ″… more important role in …″; P11 L307 …different speed, be it with… sentence looks strange.; P13 L367 Like in our previous studies… Being similar to the results of our previous studies [refs.]…)

7. Summarizing above, the whole discussion, results, and statistical sections should be re-written. At least, I do not agree with this interpretation that is not linked with your own results. Indeed, observed results (figures) were not referenced in the second, third, and fourth paragraphs of the discussion section.

8. Why does the DP, DS, and DK overlap at wide range of sped and incline? Other factors that the authors ignored could be related to the observed phenomenon, such as metabolic and muscular influences.

9. Six of 19 references (more than 30%) are based on your own studies. It looks turning everything to your own advantage. Background check could be lacking.

Reviewer #3: Synopsis:

This study aimed to isolate the effect of work rate on sub technique selection during classical style roller skiing by varying resistance force using a pulley system attached to the participants pelvis. This enabled variation of resistance force and work rate at constant speed and incline. The results show that work rate at self-selected transition between sub-technique was dependent on both speed and incline, and hence is not a direct trigger for transition. The pulley force at transition was independent of speed, but dependent on incline. Therefore, the authors conclude that resistance force might be a direct trigger for transition when incline is constant.

Assessment:

The study has a simple but clever design that apparently allows to distinguish the roles of incline, speed and work rate. It clearly adds new insight to previous studies on the same topic. However, if I interpret the results correctly, I believe the authors have overlooked the effect of gravity, which also works as a “resistance force”, when interpreting their results.

Major comment:

An important question that I think the authors do not discuss in sufficient dept is: does it matter by which mechanism resistance force is increased (e.g. by increasing incline or by adding string force close to the participant’s center of mass)? If we assume that the mechanism resistance force is added by is of minor consequence, it seems obvious that “resistance force” should be seen as the sum of string force, gravitational component along treadmill, and rolling resistance. Since the latter was constant in this experiment, and is relatively independent of incline, it can be neglected. However, the gravitational component obviously changes between the 5% and 8% incline conditions. By my calculations, the increase in resistance force due to gravity between the two inclines will be about 20 N (69 kg * 9.81 m/s^2 * [sin 4.57° - sin 2.86°]). If this is accounted for in the ANOVA (Table 1), it seems it will more than compensate the difference in resistance force that the authors report (6.7 N and 2.9 N for DP-DK and DK-DS, respectively). Based on that, I encourage the authors to reconsider how they interpret their results, and possibly modify their analysis to accommodate the change in gravitational resistance force.

Minor comments with line numbers:

- Line 14: “demand” is not very specific, consider rewording for clarity

- Line 24 and throughout: “power” and “work rate” are used interchangeably. Consider standardizing.

- Line 55: “one of these is the product of the two remaining” – this is not strictly true, suggest to either clarify or remove.

- Line 68: Missing numeric reference after Danielsen et al.

- Line 84: “In a few papers, …” Sounds very colloquial, consider rewording for conciseness.

- Line 88-90: Consider omitting the parenthesis. Check that the reference is formatted correctly.

- Line 112: use sub-technique (or gear, if you prefer) instead of “technique” for improved consistency

- Lines 117-118: Subject heights are reported using incorrect units (even Norwegians or Dutch aren’t that tall…)

- Line 133-135: consider rewording to simplify the message. How is it relevant that the subjects were familiar with the term “anaerobic threshold”?

- Line 179-181: Please consider this in light of my “major comment”

- Line 189: Consider specifying which condition these data are from (speed and incline)

- Line 202: should it really be p=0.001, or is it p<0.001?

- Table 1: include units where relevant (for force and power)

- Line 347: “However, the poling force idea makes sense”. Consider rewording, very colloquial.

- Line 396: It might not be mandatory, but I’m missing a conclusion at the very end of the manuscript.

6. PLOS authors have the option to publish the peer review history of their article (what does this mean?). If published, this will include your full peer review and any attached files.

Reviewer #1: No

Reviewer #2: No

Reviewer #3: No

---

## [Author Response · Author response to Decision Letter 0]

6 Dec 2022

We have provided all our responses point-by-point in the "Response to Reviewers" file (as one of the attached files in the submission process)

---

## [Decision Letter · Decision Letter 1]

6 Jan 2023

PONE-D-22-10768R1The role of external power on the choice of technique in classic cross-country skiingPLOS ONE

Dear Dr. Ettema,

Thank you for submitting your manuscript to PLOS ONE. After careful consideration, we feel that it has merit but does not fully meet PLOS ONE’s publication criteria as it currently stands. Therefore, we invite you to submit a revised version of the manuscript that addresses the points raised during the review process.

Please, address the minor revisions raised by reviewers 1, 2 and 3. Reviewer 4, expert in statistics, found even major issues: please, do you your best to address them too.

We look forward to receiving your revised manuscript.

Kind regards,

Luca Paolo Ardigò, Ph.D.

Academic Editor

PLOS ONE

Journal Requirements:

Additional Editor Comments:

Please, address the minor revisions raised by reviewers 1, 2 and 3. Reviewer 4, expert in statistics, found even major issues: please, do you your best to address them too.

Reviewers' comments:

Reviewer's Responses to Questions

**Comments to the Author**

1. If the authors have adequately addressed your comments raised in a previous round of review and you feel that this manuscript is now acceptable for publication, you may indicate that here to bypass the “Comments to the Author” section, enter your conflict of interest statement in the “Confidential to Editor” section, and submit your "Accept" recommendation.

Reviewer #1: All comments have been addressed

Reviewer #2: All comments have been addressed

Reviewer #3: (No Response)

Reviewer #4: (No Response)

2. Is the manuscript technically sound, and do the data support the conclusions?

Reviewer #1: Yes

Reviewer #2: Yes

Reviewer #3: Yes

Reviewer #4: No

3. Has the statistical analysis been performed appropriately and rigorously? 

Reviewer #1: I Don't Know

Reviewer #2: Yes

Reviewer #3: I Don't Know

Reviewer #4: No

4. Have the authors made all data underlying the findings in their manuscript fully available?

Reviewer #1: Yes

Reviewer #2: Yes

Reviewer #3: Yes

Reviewer #4: Yes

5. Is the manuscript presented in an intelligible fashion and written in standard English?

Reviewer #1: Yes

Reviewer #2: Yes

Reviewer #3: Yes

Reviewer #4: Yes

6. Review Comments to the Author

Reviewer #1: The authors have been very responsive to my suggestions for improving the clarity of the writing. I have no substantial issues, only minor grammatical/clarity suggestions

Title Consider adding “demand” The role of external power demand on the choice…..

Abstract

Line 24 get rid of the parentheses - …three techniques: double poling, double poling with kick and diagonal stride.

Line 28 Importantly, technique transitions were not ….

Line 32 …power) may be transition triggers when incline is fixed and speed is changed.

Line 41-42 Yet, while minimizing metabolic cost…

Line 46 …understanding the technique transitions in cross-country skiing

Line 51 DP is usually chosen on what is regarded as easy terrain, where speeds tend to be fast.

Line 66 on steep uphill terrain and at slow speeds.

Line 72 steep inclines and at slower…

Line 74 …[16,17]. However, this has been…

Line 76 …at different speeds. If one transition …

Line 76 … another one must have been at slower or faster speeds.

Line 77 suggesting that a unique…

Line 79 One approach has involved prolonged conditions…

Line 81 and record the conditions at which athletes change techniques spontaneously.

Line 96 …is supported by the finding that both at extreme …

Line 98 are chosen [19].

Line 103 … power in influencing the choice…

Line 108 cut “sub-“

Line 113 if athletes change techniques…

Line 134 provided by the laboratory

Line 137 During a ten-minute

Lines 138 & 140 cut “sub-“

Line 144 each bout (i.e. one speed-incline…

Line 145 km/hr or km hr-1

Line 146 were selected based

Line 147 expectation (singular, cut the s)

Line 158 In such case, one… (add comma)

Line 172 a custom algorithm [15]

Line 177 potential triggers (add “s”)

Line 178 … directly, the external mechanical power…

Line 179 and adding the power required to overcome the resistive force (Fp)

Line 188 cut “also”

Line 190 was also investigated as a possible trigger.

Line 190 only the mean

Line 194 during each stride cycle

Line 203 I don’t like “pulley weights” since the weight of the pulley itself is not relevant. I suggest: …showing hanging weights. C. Cycle….

Line 211 cut: “at the technique transitions”, you repeat it later in the same sentence.

Line 215 all four conditions.

Line 216 … as ‘missing value’. Yet, non-transitions can also provide valuable information. Cut: “as those cases where a transition did occur”

Line 232 cut “a”

Line 269 cut “sub-“

Line 269-270 all four conditions

Line 276 …combined transition data for increasing…

Line 279 in an opposite fashion;

Line 284 … at transition were affected by incline in opposite ways.

Line 285 … at transition were nearly identical. (not merely! �)

Line 313 This suggests that the factors that determine external power (i.e. incline, snow conditions, speed) are all important for the choice…

Line 314 cut “sub-“

Line 315 cut “sub-“

Line 315 at the steeper inclines, …

Line 316 at the shallow inclines

Line 316 equally as much

Line 319 … by incline but opposite manners, while minimally affected by speed.

Line 321 at fixed inclines, (plural)

Line 324 despite the fact that gravitational…

Line 325 at transition) comprised at least..

Line 338 is not. Comparing transitions at the same…

Line 339 at the same reistance, not the same power

Line 340 values for the speed effect on the force at transition, it should be noted…

Line 343 we cannot conclude that resistive

Line 344 where only incline was changed neither

Line 345 with the exception

Line 353 + cut: “also when not determined by speed or incline”

Line 354… be regarded as a trigger for technique transitions.

Line 355 External power is calculated from a combination of speed….

Line 360 cut “sub-“ twice!

Line 364 external power itself unlikely…

Line 364 trigger for technique transitions…

Line 371 likely triggers, though not easily measured.

Line 380 vs.

Line 389 start a new paragraph at: Moreover, we cannot…

Line 394 start a new paragraph at The current findings…

Line 398 athletes did the same

Line 401… mechanical energy, i.e. “falling on the poles”.

Line 408 all propulsive force ultimately is applied via the poles

Line 409 At the higher resistances

Line 411 become too great.

Line 413 paradigm – supports the pole force trigger hypothesis.

Line 429 we have no evidence

Line 439 not caused them.

Line 439 inherent to XC skiing.

Line 440 in this study, (add comma)

Line 449 I suggest adding: “However, based on the present study, we can now reject external mechanical work demand as an important trigger for cross-country skiing technique transitions.

Line 451 re: (visual) it might be interesting to explore if visual optic flow affects xc ski technique transitions during treadmill roller skiing. It seems to be a hot topic now according to Google Scholar. Google: “optic flow gait transition”

Reviewer #2: The manuscript was greatly updated. Most of them are good, but there are several additional requests for the authors.

1. L39; Please delete a term "as".

2. L226; Please delete a term "of".

3. L334; Replace a term "factor (single-noun)" by "factors (plural noun)".

4. L369-L370; This sentence is a repetition of the previous sentence. Please delete it.

5. L371; Please add a phrase just after the sentence like this. ... more likely triggers, which is similar to walk-run transition [1.6].

6. L410; Please replace a term "muscle force" by "muscle activities".

Reviewer #3: I appreciate the major revisions made by the authors. I think they have made the manuscript stronger. The authors have appropriately resolved my minor comments. My major comment, i.e. that gravitational force along the skiing direction changes between incline conditions, and that the appropriate outcome variable therefore should be the sum of resistance forces rather than string tension, is discussed, although not at great depth. I agree with the authors argumentation on this point, although I would have preferred to present it differently. After all, the reported aim of the study was to test if power output at technique transition is constant across different speeds and inclines. Yet, the authors also address the question if “resistance force” is constant across different speeds and inclines. This is obviously a different hypothesis, and it is not straight-forward to answer if one is keen to distinguish between incline and “resistance force” (as should be clear from the present findings and discussion). The authors may consider again if they think the manuscript benefits from this discussion, or if it is better to keep it closer to the stated aim (i.e. power). Since I find nothing formally wrong in their arguments, this should be the authors’ prerogative.

Minor comments:

- Line 111: missing word: “The comparison four such conditions…”

- Lines 214-219: Is imputing missing values in this way appropriate?

- Figure 2 is difficult to understand, and (if I understand it right), it is not very important. Are you sure you want to initiate the results-section with this figure, and can it be presented/explained better?

- Lines 253-254 (or perhaps elsewhere, but this is where I thought about it): When interpreting effects of speed and incline, perhaps it’s worth reflecting on that the change in power output (at equivalent string force) between inclines was much bigger than the change in power output between speeds? By my calculations, power output during the two incline conditions differed by 44% while the speed conditions differed by 20%.

- Lines 299-300: “i.e., the athletes retained a technique at greater resistive force when resistance was incremented versus decremented”. Do you mean power instead of force?

- Lines 392-394: I’m not convinced by the statement “Still, the fact that additional force and total force (and power) at transition show opposite effects of incline (Fig. 4) suggests that these variables are essentially different.” Can you elaborate?

- Lines 394-396: Is the reference to Pellegrini et al. appropriate, given that they measured pole force while you measured (or estimated) total resistance force?

- Lines 455-460: This seems unnecessary to mention in a conclusion.

- Overall: Font size changes a few times in the manuscript body

Reviewer #4: The revision is not much improved, statistically, from the original submission. The results may still hold, but not in a statistically rigorous manner. As pointed out by a previous Reviewer, this is probably due to a lack of appropriate number of participants. The statistics and result sections are confusing. There is no design justification, power or sample size considerations motivating the small size of this investigation The test of normality for ANOVA is not adequately detailed. Reviewer 2, comment 4.1. is still relevant. The total of 14 is still much too inadequate considering the multiple endpoints and multiple comparison issues-especially with the p-value interpretation in the classical setting. Even for a Bayesian approach , the lack of sample size challenges the credibility and interpretation of the results.

The investigators note that they have a more holistic approach in presenting both the classical and Bayesian results (rather than solely focusing on ‘significance’) thus they can reduce the chance of committing type I errors. Again, one needs adequate sample size to reduce the chance of committing a type one error.

This holistic argument (classical and Bayesian) is not really justified. This confuses the results for the reader and one should use one or the other and present the results clearly.

The concept of missing value accommodation is not clear at all. The investigators note that (page 10 to 11) to allow them to use such ‘missing value’ in the statistical analysis, it was filled in with a fictitious value, one force or power step higher than the maximal value or one step less than the minimal value observed for that athlete during all four incline-speed settings. What is the reference for this approach?

Regarding the Bayesian approach, what underlying distributions are used for the prior constructs? How are the prior constructs arrived at? Presumably the approach is a Markov chain Markov modeling technique with diffuse or vague priors. All of the actual mechanics of this technique in this context is missing.

7. PLOS authors have the option to publish the peer review history of their article (what does this mean?). If published, this will include your full peer review and any attached files.

Reviewer #1: No

Reviewer #2: No

Reviewer #3: No

Reviewer #4: No

---

## [Author Response · Author response to Decision Letter 1]

6 Feb 2023

Please see attached Reply-to-Comments file

---

## [Decision Letter · Decision Letter 2]

3 Mar 2023

PONE-D-22-10768R2The role of external power demand on the choice of technique in classic cross-country skiingPLOS ONE

Dear Dr. Ettema,

Thank you for submitting your manuscript to PLOS ONE. After careful consideration, we feel that it has merit but does not fully meet PLOS ONE’s publication criteria as it currently stands. Therefore, we invite you to submit a revised version of the manuscript that addresses the points raised during the review process.

Please, one further effort to address Reviewer 1's minor issues. 

We look forward to receiving your revised manuscript.

Kind regards,

Luca Paolo Ardigò, Ph.D.

Academic Editor

PLOS ONE

Journal Requirements:

Additional Editor Comments :

Please, one further effort to address Reviewer 1's minor issues.

Reviewers' comments:

Reviewer's Responses to Questions

**Comments to the Author**

1. If the authors have adequately addressed your comments raised in a previous round of review and you feel that this manuscript is now acceptable for publication, you may indicate that here to bypass the “Comments to the Author” section, enter your conflict of interest statement in the “Confidential to Editor” section, and submit your "Accept" recommendation.

Reviewer #1: All comments have been addressed

Reviewer #2: All comments have been addressed

Reviewer #3: All comments have been addressed

Reviewer #4: All comments have been addressed

2. Is the manuscript technically sound, and do the data support the conclusions?

Reviewer #1: Yes

Reviewer #2: Partly

Reviewer #3: Yes

Reviewer #4: (No Response)

3. Has the statistical analysis been performed appropriately and rigorously? 

Reviewer #1: I Don't Know

Reviewer #2: Yes

Reviewer #3: I Don't Know

Reviewer #4: (No Response)

4. Have the authors made all data underlying the findings in their manuscript fully available?

Reviewer #1: Yes

Reviewer #2: (No Response)

Reviewer #3: Yes

Reviewer #4: (No Response)

5. Is the manuscript presented in an intelligible fashion and written in standard English?

Reviewer #1: No

Reviewer #2: Yes

Reviewer #3: Yes

Reviewer #4: (No Response)

6. Review Comments to the Author

Reviewer #1: The authors have worked hard to improve the English writing/grammar.

I applaud their persistence.

It was a pleasure to read the revised manuscript.

A few minor grammar items that remain can be easily fixed.

Line 99 too many negatives in one sentence. Simpler structure: “Still, in all these experiments changes in power were always associated with changes in incline and/or speed and so the isolated role of power was not clear.”

Line 147-148 … assuring that they would be submaximal efforts, and the expectation…. (this fix avoids confusion between “load” and the hanging weights which might be called “loads”)

Line 187 … to investigate the role of power, it was also possible to elucidate the means by which power was altered, i.e. the resistive force.

Line 194 …external mechanical power was examined; any fluctuations in power due to the motion…..

Line 196 …associate losses were not considered.

Line 216 … conditions was regarded as evidence

Line 323 … but in opposite

Line 398 independent of the way

Lines 400 and 402 I’ve never heard this phrase ‘sense for propulsion’ but it would make more sense in English if it was ‘sense of propulsion’. More common would be ‘sense of effort’

Line 458 i.e. smaller weight increments (reducing could be confused with when your experiment removed weight bags)

Line 463 I suggest you avoid two uses of the word “trigger”. Here, simply say: “This raises the question …”

Line 467 “Still and anyway” is not something I’ve ever read or said. Instead, try: “In any case, the current findings….”

Line 481 statistical

Line 481-482 I don’t understand what is meant here. Perhaps one of the other, more statistical reviewers can clarify.

Reviewer #2: (No Response)

Reviewer #3: (No Response)

Reviewer #4: The authors did address the concerns for the holistic approach. However, it is still confusing to a reader expecting a consistent explanation for the statistical results.

7. PLOS authors have the option to publish the peer review history of their article (what does this mean?). If published, this will include your full peer review and any attached files.

Reviewer #1: No

Reviewer #2: No

Reviewer #3: No

Reviewer #4: No

---

## [Decision Letter · Decision Letter 3]

20 Mar 2023

The role of external power demand on the choice of technique in classic cross-country skiing

PONE-D-22-10768R3

Dear Dr. Ettema,

We’re pleased to inform you that your manuscript has been judged scientifically suitable for publication and will be formally accepted for publication once it meets all outstanding technical requirements.

Kind regards,

Luca Paolo Ardigò, Ph.D.

Academic Editor

PLOS ONE

Additional Editor Comments (optional):

Congratulations for the interesting work.

Reviewers' comments:

Reviewer's Responses to Questions

**Comments to the Author**

1. If the authors have adequately addressed your comments raised in a previous round of review and you feel that this manuscript is now acceptable for publication, you may indicate that here to bypass the “Comments to the Author” section, enter your conflict of interest statement in the “Confidential to Editor” section, and submit your "Accept" recommendation.

Reviewer #1: All comments have been addressed

2. Is the manuscript technically sound, and do the data support the conclusions?

Reviewer #1: Yes

3. Has the statistical analysis been performed appropriately and rigorously? 

Reviewer #1: Yes

4. Have the authors made all data underlying the findings in their manuscript fully available?

Reviewer #1: Yes

5. Is the manuscript presented in an intelligible fashion and written in standard English?

Reviewer #1: Yes

6. Review Comments to the Author

Reviewer #1: In my suggestion, I made a typographical error.

line 196 centre of mass, mechanical energy fluctuations of the body and associate losses were not...

should read "associated"

Otherwise it's fine! and hopefully it will be published by the NEXT cross-country ski season.

7. PLOS authors have the option to publish the peer review history of their article (what does this mean?). If published, this will include your full peer review and any attached files.

Reviewer #1: No

---

## [Editor Report · Acceptance letter]

23 Mar 2023

PONE-D-22-10768R3 

The role of external power demand on the choice of technique in classic cross-country skiing 

Dear Dr. Ettema:

I'm pleased to inform you that your manuscript has been deemed suitable for publication in PLOS ONE. Congratulations! Your manuscript is now with our production department. 

Kind regards, 

on behalf of

Dr. Luca Paolo Ardigò 

Academic Editor

PLOS ONE